# Remanufacturing Decision-Making for Gas Insulated Switchgear with Remaining Useful Life Prediction

**Seokho Moon** [1], **Hansam Cho** [1], **Eunji Koh** [1], **Yong Sung Cho** [2], **Hyoung Lok Oh** [3], **Younghoon Kim** [4,*] **and Seoung Bum Kim** [1,*]

1 School of Industrial and Management Engineering, Korea University, 145 Anamro, Seongbuk-gu, Seoul 02841, Korea
2 Advanced Power Apparatus Research Center, Korea Electrotechnology Research Institute, 12, Jeongiui-gil, Seongsan-gu, Changwon-si 51543, Korea
3 WithBeer Co., Ltd., Industry Research Center, 50, Hyeoksinsandan 1-gil, Naju-si 58277, Korea
4 Department of Industrial and Management Systems Engineering, Kyung Hee University, 1732, Deogyeong-daero, Giheung-gu, Yongin-si 17104, Korea
* Correspondence: y.kim@khu.ac.kr (Y.K.); sbkim1@korea.ac.kr (S.B.K.)

**Abstract:** Remanufacturing has emerged as a way to solve production problems, as raw material costs increase and environmental pollution caused by discarded equipment occurs. The process can extend product lifetime and prevent waste of resources. In particular, it has economical efficiency for large equipment such as GIS (Gas Insulated Switchgear). The crucial points in remanufacturing are determining replaceable parts and economic valuation. To address these issues, we propose a framework for remanufacturing GIS with remaining lifetime prediction. We construct a regression model for remaining useful life (RUL) in the proposed framework using GIS sensor data. The cost of the replacement parts is estimated with the selected sensors. To validate the effectiveness of the proposed framework, we conducted accelerated life testing on a GIS for data acquisition and applied our framework. The experimental results demonstrate that the tree-based RUL regression model outperforms the others in prediction accuracy. In the simulation of part replacement, the important sensor-based decision-making improves RUL significantly.

**Keywords:** remanufacturing; gas-insulated switchgear; remaining useful life regression; accelerated life testing; replacement simulation

## 1. Introduction

Reliable power supply for metropolitan areas has become increasingly crucial as urbanization has progressed [1]. Substations play a vital role in reliable power supply by safely converting power from a power plant located outside of cities to power for urban homes. Gas Insulated Switchgear (GIS) is one of the essential pieces of equipment in the substation, which isolates dangerous high-voltage current from external contact using sulfur hexafluoride ($SF_6$) gas [2]. In addition, when a fault current occurs, a circuit breaker in GIS can instantly cut off the current.

In the 1970s, a large number of GISs were installed in response to a rapid increase in electricity demand in the Republic of Korea. Most GISs have recently exceeded their design life; thus, the demand for a new product or broken parts replacement dramatically increased. However, the production of new GIS equipment entails considerable costs due to a rapid rise in raw material prices. The destruction process of GIS also releases harmful gases, leading to environmental pollution [3]. Moreover, multinational corporations should follow regulations about eco-friendly management [4].

Therefore, remanufacturing has been proposed as one of the reasonable solutions to these problems. Remanufacturing is reusing parts that can be utilized in existing equipment as-is, and replacing damaged parts [5]. This method is cost effective because it

can significantly extend the product's life and minimize the energy required for assembly [6]. Industry-specific criteria for remanufacturing differ. In the case of a grinding machine, for instance, the remanufacturing criteria include economic feasibility, technical feasibility, and resource environment feasibility [7]. In this paper, the criteria for remanufacturing in GIS are an improvement in product life and the economic feasibility of part replacement.

The expected improvement in product life after GIS parts replacement can be estimated using RUL analysis, which has been utilized to inspect and monitor equipment conditions in various industries, including automobiles, engines, and electronic devices [8]. The RUL prediction is classified as knowledge based, data driven, and hybrid [9]. Data-driven RUL prediction employs multiple regression models of machine learning algorithms and is applied to the collected signal data. Based on the correlation between the input sensor data and the RUL, the models can be constructed without expert knowledge. Deep learning models with huge amounts of training data have demonstrated superior performance [10]. In the real-world data from manufacturing factories, tree-based models are generally employed because these models can calculate feature importance based on the relationship between input and output data, as indicated by the amount of impurity reduction, and can interpret the prediction results [11]. In this study, we derive an expected RUL after GIS parts replacement using the RUL regression model based on sensor data.

There are several studies for the GIS equipment based on sensor data. However, most researchers have focused on fault detection. One of the significant faults in GIS is partial discharge causing tremendous damage to the entire substation [12,13]. These studies use ultra-high-frequency sensors to measure electromagnetic waves from 300 MHz to 3 GHz generated by partial discharge. Some studies for real-time fault diagnosis of the high-voltage circuit breaker in GIS are proposed using the current waveform during open/close operations [14,15]. These researchers construct fault detection models of Support Vector Machine, Kernel Principal Component Analysis Random Forest, and Autoencoder. Likewise, latent inner insulation defects could be detected from analysis of $SF_6$ decomposition using photoacoustic spectroscopy gas sensors [16].

A few studies have utilized signal data to detect the degradation of GIS. To evaluate an optimal design robust to insulation degradation, several cases of voltage and insulation thickness configuration were examined [17]. A limitation of this research is that they only used a voltage sensor. Zhang et al. proposed a remaining lifetime estimation method using sensor parameters responsive to degradation [18]. They demonstrated a failure rate function that can calculate a remaining lifetime according to Weibull distribution. However, this method required expert knowledge on a circuit breaker in detail. Moreover, when the entire data of this study change, it is hard to determine whether modeling based on the Weibull distribution is valid or not.

In this study, we acquire and utilize seven signal data from each position relative to GIS failure. Next, RUL regression models based on six machine learning algorithms are constructed to estimate the remaining useful life. Finally, we propose a process suggesting the most reasonable replacement parts. The following is a summary of this study's main contributions:

- To the best of our knowledge, this is the first attempt to propose a data-driven remanufacturing decision-making framework in GIS equipment. With the test R-squared 0.999, the RUL regression model shows state-of-the-art prediction performance.
- We acquire signal data from GIS with accelerated life testing by setting up a laboratory. The data consist of seven signal data crucially related to the degradation of GIS.
- The replacement simulation confirms that the proposed framework is valid for improving RUL economically.

## 2. Proposed Framework

The proposed remanufacturing decision-making consists of two steps, as shown in Figure 1. In the first step, we acquire sensor signal data from GIS, then construct an RUL regression model. To acquire data, we attach several sensors to main positions of GIS.

Each collected datum from sensors is subjected to a preprocessing procedure that includes noise elimination and time windowing. The preprocessing is performed according to the frequency of each sensor. For the regression model, the inputs are sensor data and the output is the RUL. This framework compares six machine learning models. After training, the best-performing model is determined for a sensor replacement simulation.

**[ Step 1 : Data collection & RUL regression ]**

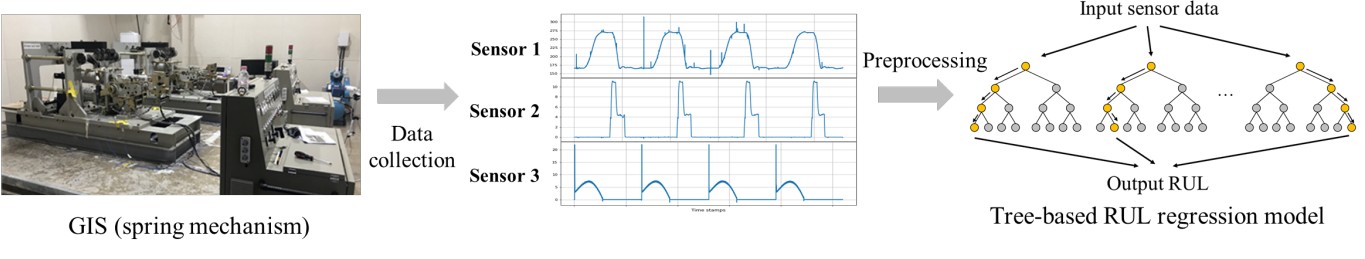

**[ Step 2 : Sensor replacement simulation & Remanufacturing  decision-making ]**

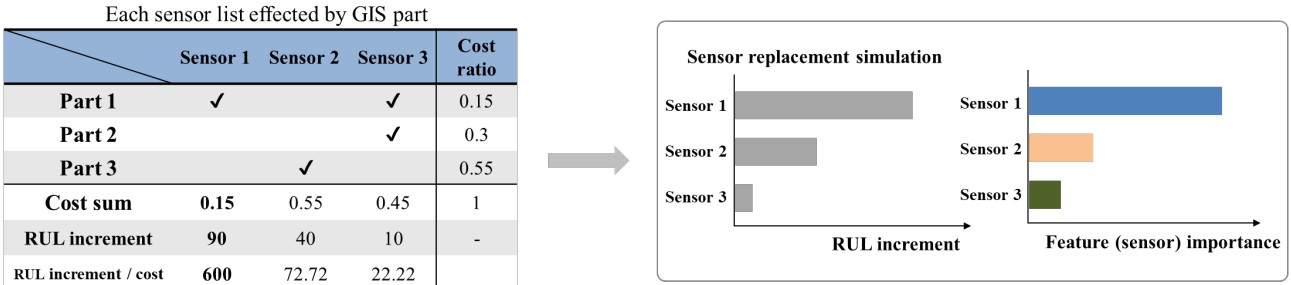

**Figure 1.** Overall decision-making framework for data-driven remanufacturing .

In the second step, the sensor replacement simulation is performed with the optimal model. We can calculate RUL increment and RUL increment per cost based on the model. The ratio of each cost to the total cost is calculated instead of using an actual cost, because of confidentiality issues. We set the increment of RUL per cost for each sensor as a remanufacturing criterion. Lastly, we validate a relation between the sensor importance and the increment of RUL. Thus, the framework provides a priority list of sensors that are the most cost effective after replacement.

*2.1. Data Collection*

The GIS comprises insulators, manipulators, actuators, a circuit breaker, and a frame. The circuit breaker can stop the circuit so that the fault current does not affect the other power equipment when a short circuit accident occurs. In accordance with the type of force that cuts the circuit, there are three categories: pneumatic mechanism, hydraulic mechanism, and spring mechanism. A procedure of circuit breaking has three operations, as shown below: (1) converting electrical signals to mechanical signals; (2) producing operating force by actuating valves (pneumatic or hydraulic) or springs; (3) applying the operating force to the circuit breaker and link. This work uses the GIS with a spring mechanism circuit breaker, because most aging GIS was manufactured with this circuit breaker. This equipment includes a hook, motor, open/close shaft, springs, and limit switch. Each part is used in the circuit-breaking procedure (1), (2), or (3) described above.

To validate the data-driven remanufacturing decision making, we set up accelerated life testing (ALT) on GIS with a spring mechanism circuit breaker. Due to limited resources and time, we acquired sensor data from the circuit breaker, which is the most crucial GIS part for degradation. The dataset was collected from our laboratory because this research is the first attempt to construct a new framework on GIS equipment. ALT is necessary to acquire the dataset for RUL regression because the life of GIS is between 30 and 50 years. ALT is one of the most well-known methods of reducing the time of data acquisition by exposing equipment to an extreme environment [19,20]. These extreme environments usually are obtained by increasing temperature, pressure, humidity, voltage, or the number of operations to be harsher than the normal usage conditions [21]. In our case, we constructed that the number of operations is main stress of ALT. Until the GIS was broken, the ALT had been conducted for five months. We use one type of 170 kV 50 kA 60 Hz specification GIS with a spring mechanism circuit breaker. Based on a GIS Failure Modes and Effects Analysis (FMEA) result, each sensor is attached to a position highly related to failure during open/close operations. FEMA is usually used as a guideline for reducing the occurrence of failures in equipment parts design [22].

Table 1 shows the sensor list, including sensing position, value, and type. There are seven sensor positions, and three types of sensor devices exist. The overall process of data flow from stroke data is shown in Figure 2. The Stroke measurement consists of contact type (rotary sensor) and non-contact (laser distance sensor). As shown in Figure 3a, we used a laser distance (LD) sensor to get stroke information. The LD sensor is generally used for non-contact distance measurement by controlling laser beams [23]. The measuring position of the LD sensor is an open spring compression plate where a linear reciprocating motion occurs. This sensor has high durability and can minimize vibrations of the sensor caused by open/close operations. Hence, this sensor can measure more accurately and stably than contact-type sensors. As shown in Figure 3b, every current from the open/close trip coil and the auxiliary coil was collected via direct current transformer (DC CT) sensors. A current transformer (CT) sensor is widely used to measure high current in alternating current. In particular, the DC CT is a sensor developed to measure current in direct current. In our experiment, the measured current data are used to verify a trip speed and damping level after contact points are opened. The auxiliary contact signal of the open/close state can be acquired from a mechanical auxiliary contact, and we digitized this contact information. By measuring the motor current using DC CT, it is possible to monitor the state of the motor. In addition, we can analyze whether the motor overheated through the motor temperature sensor.

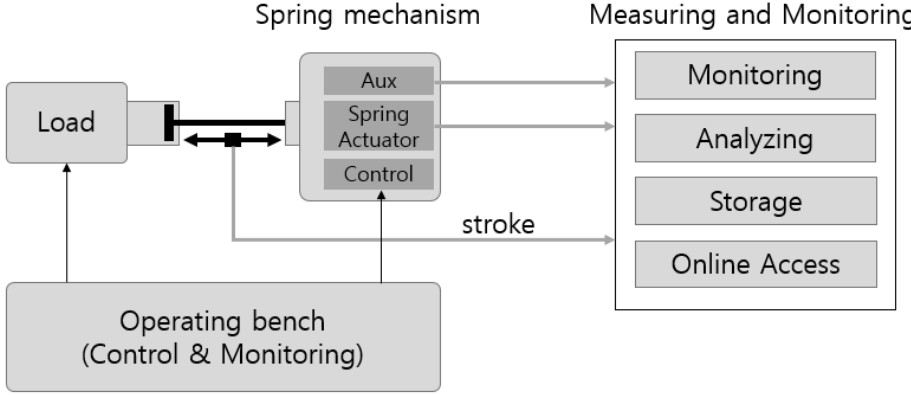

**Figure 2.** The illustration of stroke sensor data flow processing on spring mechanism GIS.

**Table 1.** Data description of each sensor.

| No. | Sensing Position | Sensing Value | Sensor Type |
|---|---|---|---|
| 1 | Simulated load | Distance | LD sensor |
| 2 | Open trip coil | Current | DC CT |
| 3 | Close trip coil | Current | DC CT |
| 4 | Auxiliary coil | Current | DC CT |
| 5 | Auxiliary contact | Contact signal | - |
| 6 | Motor | current | DC CT |
| 7 | Motor | temperature | Temperature detector |

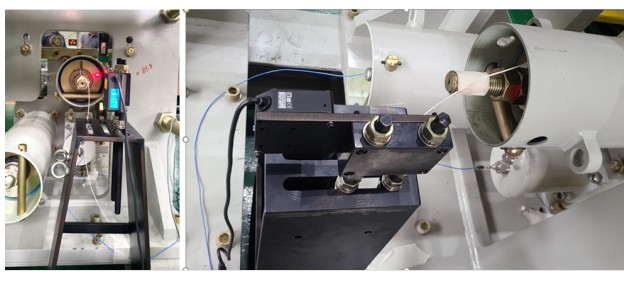 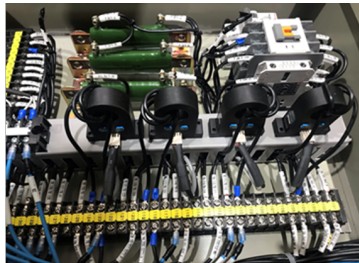

(a) LD (Laser Distance) Sensor       (b) DC CT Sensor

**Figure 3.** (**a**) LD sensor for measuring stroke distance. (**b**) DC CT sensor for measuring current of open/close trip coil and auxiliary coil.

## 2.2. Data Preprocessing

To train the RUL regression model, we preprocessed our data in a proper form. The entire preprocessing process has three steps, as stated below: (1) defining input data, output data, and each observation; (2) splitting data into train, valid, and test sets; (3) data normalization.

First, input data consist of six signals, except for the auxiliary contact signal. The auxiliary contact signal is excluded because the information only verifies whether the open/close operation works well. In our experiment, we use RUL as output data. RUL is generally calculated based on the number of operations until a machine fails. To calculate the RUL of a lithium-ion battery, for instance, the number of charging and discharging cycles until failure is required [24]. Likewise, the number of open/close operations until failure is necessary for calculating the RUL of GIS. In this paper, $M$ is defined as the maximum number of open/close operations before the GIS fails. RUL can be calculated based on the following assumption. Our study assumes that the RUL and the cumulative number of open/close operations have a linear relation, as shown in Figure 4. Therefore, the RUL of GIS decreases proportionately to the cumulative number of operations. GIS manufacturing experts confirmed our assumption of the GIS degradation process. Thus, the RUL can be defined as:

$$RUL_k = (1 - \frac{k}{M}) \times 100, \text{ for } k = 1, 2, 3, ..., M \tag{1}$$

$k$ is the cumulative number of open/close operations.

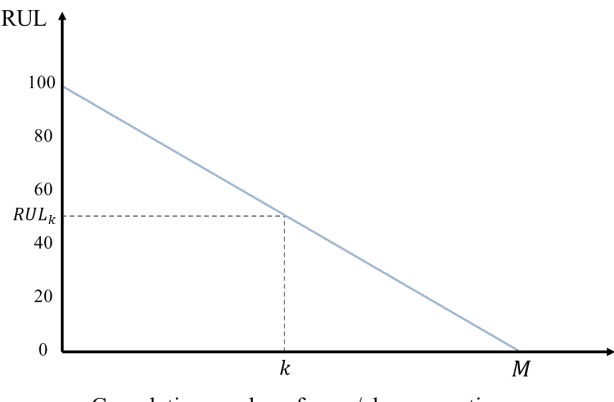

**Figure 4.** RUL linear function of GIS equipment.

We define an observation consisting of input and output data during each open/close operation. The total number of observations is 14,359, equal to the maximum number of operations. For each observation, all signal data except motor current follow 10,000 Hz and have a 0.28 s duration. Motor current signal data follow 1000 Hz and have 26 s duration. In addition, there is a time interval of 150 s between every observation, as shown in Figure 5, because ALT needs to have the time interval for its stability despite time inefficiency. To handle the issue of the different sensors' duration, signal data from each observation is equally divided into ten splits. In every split, we extracted seven types of statistics, including mean, standard deviation, minimum, first quartile, second quartile (median), third quartile, and maximum. Figure 6 shows examples of ten splits of each sensor. Therefore, 70 statistics are extracted from ten splits, so one observation has 420 statistics as features.

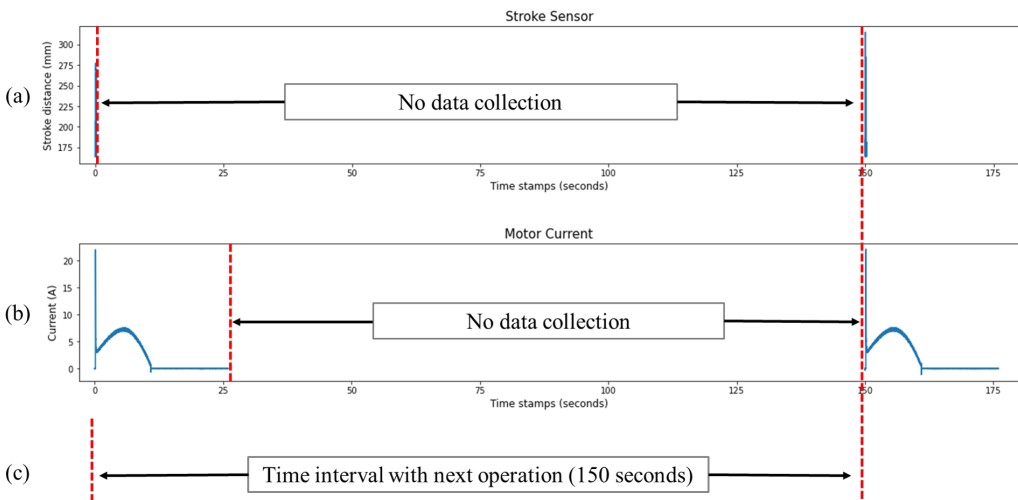

**Figure 5.** (**a**) is a sampled signal datum using 10,000 Hz sensor and represents 0.28 s duration per each operation. The stroke sensor is used as an example of visualization (**a**), and the other sensors, including trip coil, auxiliary, and motor temperature, also have the same frequency and duration. (**b**) is a sampled signal datum from motor sensor using a 1000 Hz sensor and represents 26 s duration per each operation. (**c**) is a visualization of time interval between operations.

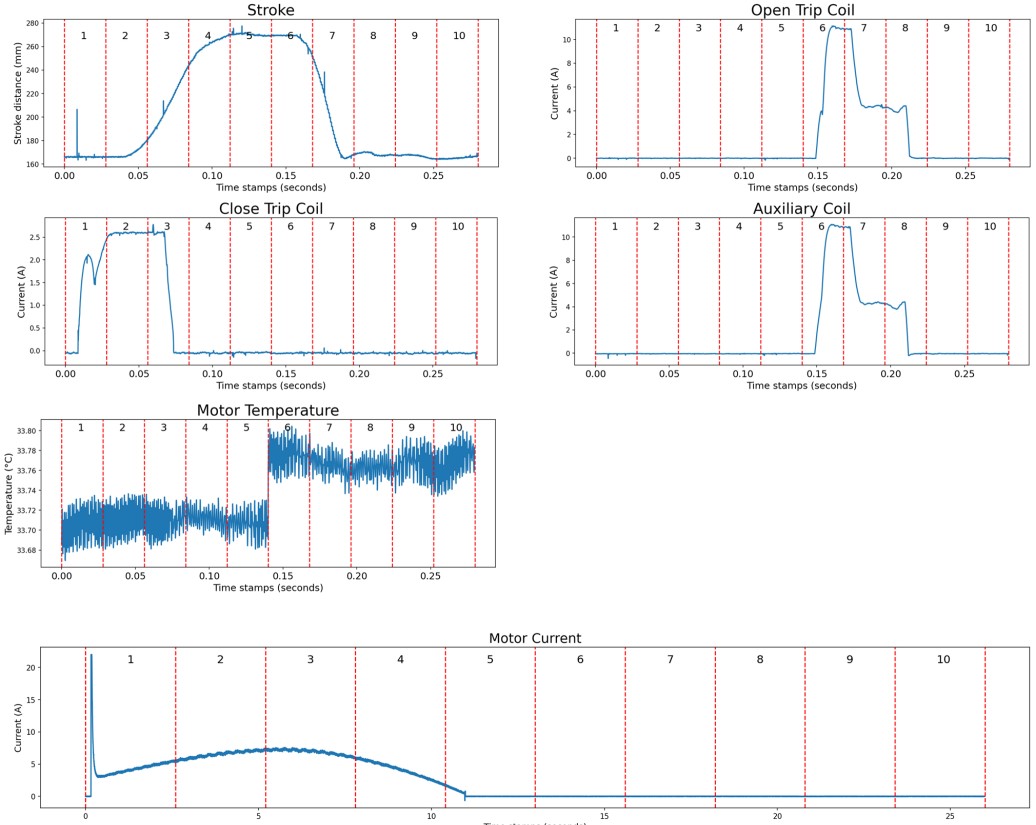

**Figure 6.** Visualization of splitting sensor signal data into ten to extract statistics during one open/close operation.

Next, we partitioned the entire dataset into training, validation, and testing data. Before extracting statistics, noises of input sensor data were reduced by employing the moving average smoothing method. This method is commonly used for noise reduction in time series data [25]. In this experiment, we configured that a window length is 13. This means that the values of each data point are changed to the average value of the previous 13 data points. To evaluate RUL regression model, 60% of the data were used for training, 20% for validation, and the remaining 20% for testing. Then, we conducted normalization on an extracted statistic from input data. Standard scaling was employed for preprocessing input data. Standard scaling is a method in which input data are transformed to be centered around a mean with a unit of standard deviation. In this experiment, we established that means equal to zero and standard deviation is equal to one.

### 2.3. Remaining Useful Life Regression

We compare six regression algorithms (linear, Ridge, least absolute shrinkage and selection operator (LASSO), Elastic Net, Random Forest, and extreme gradient boost (XGBoost)) for application to GIS data. The prediction results suggest the best model that can find meaningful patterns between RUL and input signal. In addition, sensor importance is derived from the six models to interpret results. More details are stated as follows.

Linear regression is fundamental and widely employed for regression analysis. The least square estimates coefficients of a linear equation. Linear regression may underperform because of the overfitting to the training data [26]. Therefore, various regularization techniques have been proposed to avoid overfitting. Ridge regression simultaneously minimizes a sum of squared error and squared coefficient [27]. Adding an L2 penalty in basic linear regression can alleviate variance and achieve superior performance. The LASSO regression uses the coefficients' absolute value to regularize the linear model [28].

Unlike Ridge regression, LASSO can select features and reduce the dimensionality of data. Elastic Net combines both the L1 and L2 penalty to improve performance than a model with either one or the other penalty [29]. This algorithm can select important features and mitigate multicollinearity. Therefore, the Elastic Net is generally preferred over Lasso regression when several features are highly correlated.

Random Forest is an ensemble algorithm trained with a large number of decision trees [30]. This method is often employed for classification and regression. In regression, the final predicted value is determined by the average of each tree's predicted values. Random Forest uses bootstrap aggregation (bagging), which reduces variance. Consequently, this model handles large amounts of data effectively and improves model performance by addressing the overfitting issue. XGBoost is another ensemble algorithm based on trees. Gradient boosting is used to iteratively construct this model from weak decision tree models to make it robust [31]. This technique can predict the residuals of previous models, and predicted values are combined to produce the final result.

### *2.4. Gis Parts Replacement Simulation*

When the performance of the best regression model is superior to a certain threshold, we consider this model as a base of replacement simulation. Then, the model calculates the sensor's importance. To determine the quantitative criterion of remanufacturing, first, we simulate that each sensor datum is exchanged with a new one. Second, we compute an increment of RUL when stimulation occurs, as shown in Figure 7. One sensor data are modified to an initial value and utilized as the input data to compute the increment. Then, the RUL increment is stated as the difference between the RUL value before and after changing the sensor data. Thus, RUL increments are obtained for all sensors. We can verify whether a tendency of an RUL increment and sensor importance has a common property. Finally, we calculated an increment of RUL per cost by each sensor based on the cost ratios of each GIS part.

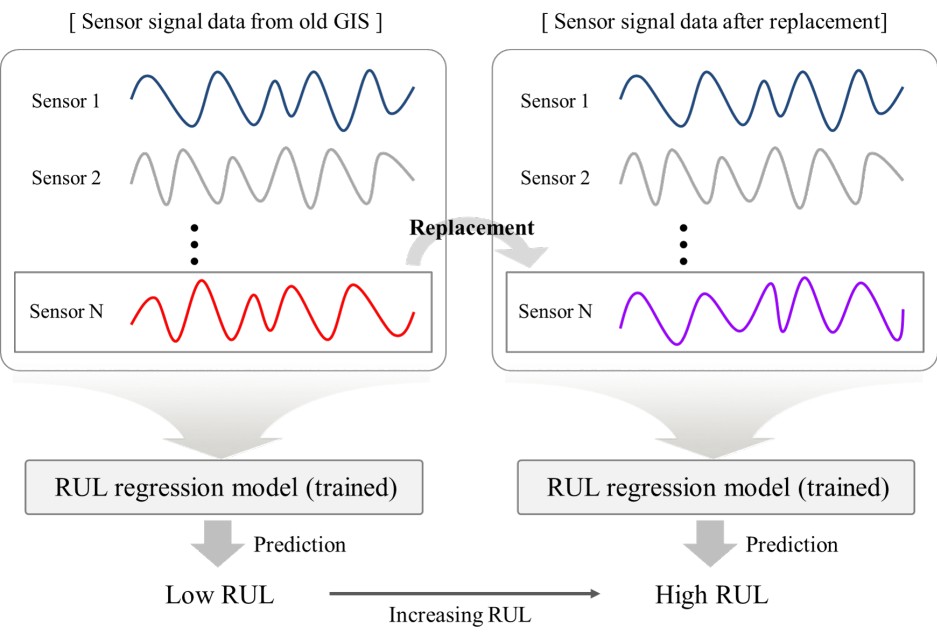

**Figure 7.** Illustration of GIS Parts replacement simulation and RUL increment.

### 3. Results

We compare the RUL regression results to evaluate which of the six models produces the best performance. Mean square error (MSE), mean absolute error (MAE), and R-Squared ($R^2$) are frequently employed for regression performance comparison. MSE is one of the most common metrics, which is also used for model training loss. MAE can confirm a mean of every residual between predicted RUL and actual RUL by model. $R^2$ in the regression

model is an indicator that shows how well the input features explain the output feature. Cross-validation was conducted 20 times to verify the experimental results' consistency, and we reported the averaged values. Key performance indicators (KPI) of MSE and $R^2$ were set to 2 and 0.95, respectively. As reported in Table 2, Random Forest outperformed the other machine learning models in terms of all metrics, including MSE, MAE, and $R^2$, and was set to a base model for the replacement simulation.

**Table 2.** Comparisons of results using six regression algorithms, in terms of MSE, MAE, and R-Squared. The best performances are in bold and the mean and standard deviations are reported.

| Model | MSE | MAE | $R^2$ |
|---|---|---|---|
| Linear Regression | 46.824 (53.815) | 2.324 (0.101) | 0.940 (0.068) |
| Ridge | 32.196 (40.842) | 2.351 (0.091) | 0.959 (0.051) |
| Lasso | 87.764 (1.789) | 7.519 (0.091) | 0.887(0.002) |
| Elastic Net | 85.772 (1.924) | 7.433 (0.101) | 0.890 (0.003) |
| Random Forest | **0.463 (0.143)** | **0.263 (0.014)** | **0.999 (0.000)** |
| XGBoost | 0.765 (0.176) | 0.509 (0.018) | 0.999 (0.000) |

Table 3 presents a list of GIS parts which are related to each sensor. For instance, the stroke sensor is affected by GIS parts, including the close shaft, open shaft, dashpot, and link. To change stroke sensor data to initial values, we should replace the four parts with new ones. The cost of replacement is the sum of costs replacing every part affecting the corresponding sensor. For example, the cost ratio of replacement stroke sensor is 0.529 (=0.225 + 0.215 + 0.075 + 0.014) from Table 3. Note that the cost in dollars is confidential information, so we report the proportion of the cost of parts against the whole parts.

**Table 3.** List of parts that affect each sensor data. The RUL increment per cost is reported. ✓ mark means sensor and corresponding parts are closely related.

| Parts \ Sensor | Stroke | Open Trip Coil | Close Trip Coil | Auxiliary Coil | Motor Temperature | Motor Current | Cost Ratio |
|---|---|---|---|---|---|---|---|
| Gear | | | | | ✓ | ✓ | 0.135 |
| Open/close hook | | ✓ | ✓ | ✓ | | | 0.125 |
| Close shaft/spring | ✓ | | | | ✓ | ✓ | 0.225 |
| Open shaft/spring | ✓ | | | | | | 0.215 |
| Dashpot | ✓ | | | | | | 0.075 |
| Link | ✓ | | | | | | 0.014 |
| Frame | | | | | ✓ | | 0.211 |
| **Sum cost** | 0.529 | 0.125 | 0.125 | 0.125 | 0.571 | 0.36 | 1 |
| **RUL increment** | 24.09 | 0.11 | 0.02 | 0.23 | 0.12 | 96.88 | - |
| **RUL increment per cost** | 45.53 | 0.88 | 0.16 | 1.84 | 0.21 | **269.11** | - |

Table 3 also shows the RUL increment per cost, which is calculated by dividing the RUL increment by the cost ratio. The higher value means the corresponding sensor is more cost effective to replace than others. The result verifies that the motor current sensor is the most cost effective because an RUL increment per cost is 269.11. The value is greater than the second-best value by approximately six times.

We depicted the importance of each sensor in terms of the Random Forest, as shown in Figure 8. The values are computed as the mean of accumulation of the impurity decrease within each tree. The motor current sensor is considerably more important than other sensors, and the stroke sensor is the second most important sensor. The other sensors show insignificant values. The RUL increment for each sensor is specified in Table 3. Interestingly, we confirmed that RUL increment and sensor importance are highly correlated as shown in Figure 9. The result implies that the Random Forest precisely found out the patterns of relations between RUL and sensor inputs.

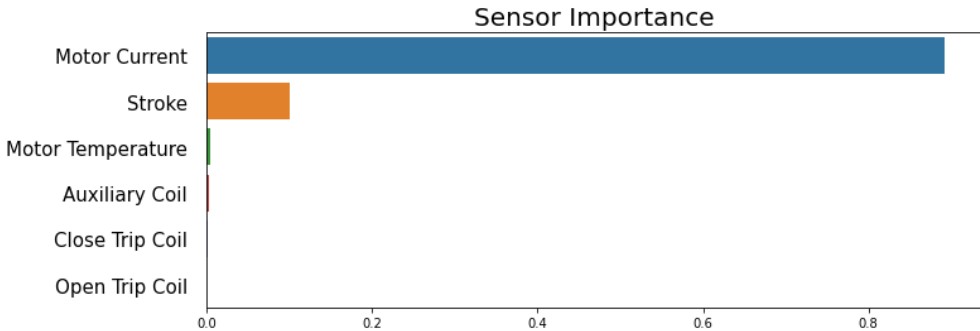

**Figure 8.** Feature importance sorted by descending order from the Random Forest.

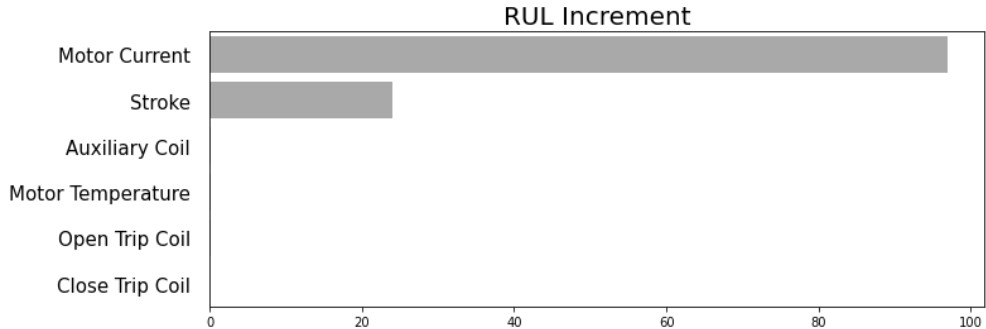

**Figure 9.** RUL increment sorted by descending order from the Random Forest.

## 4. Discussion

The sensor signal is a type of time series data, and it is common to use models extracting sequential features. For instance, there are traditional ARIMA, deep-learning-based Long Short-Term Memory (LSTM), and Transformer [32] models. Nonetheless, we used multivariate analysis models. In the dataset, there is an interval where no data are collected between each open/close operation during the ALT. Since we assumed one open/close operation as one observation, every observation may have a few time series features. To evaluate whether the input data are suitable for time series models, we constructed an LSTM model for RUL regression using raw signal data. LSTM is based on a structure of a recurrent neural network that learns continuous information and is robust to information loss. The model is one of the deep learning algorithms mainly used for handling signal data [33]. However, the LSTM model performed poorly in our validation process, so it could not be used in the proposed framework ($R^2$ = 0.105). We guessed that the signal length of each observation is quite long, so it may have been difficult for the LSTM model to extract features properly. Another conjecture was as follows. As shown in Figure 10, statistics from each split might be more meaningful information for predicting the number of open/close operations than continuous features such as the shape of the waveform. The motor current value used for statistics appears to be different, even though the waveform is nearly the same as the number of operations increases.

We excluded the time series models for RUL regression for the same reason. Thus, we constructed six RUL regression models that use the seven statistics extracted from each observation as input values. The tree-based models achieved the best performance and explainability among the six regression models.

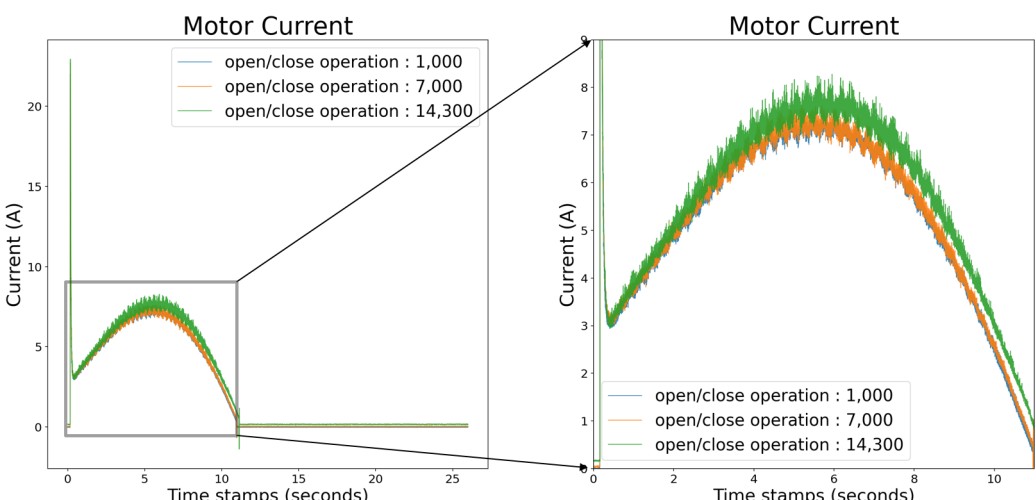

**Figure 10.** (**left**) A graph of the corresponding motor current values when 1000, 7000, and 14,300 open/close operations are conducted. (**right**) Zooming into the 0 to 11 second range and the 0 to 9 motor current value range.

An ability to interpret a model's result is particularly crucial in a real-world industrial application such as remanufacturing decision making. It is possible to confirm which sensor affects degradation quantitatively, and this can be easily seen via visualization. Motor current, for example, is the most significant sensor, and can be classified as the top priority using post-analysis. Figure 10 depicts the current value of the motor is comparable near the start of the experiment (1000) and the middle (7000) of the total number of operations. However, the current value near the experiment's start (1000) differs near the end of the experiment (14,300). In detail, the highest current value can be observed just before the failure. According to the expert's assessment, two possible explanations exist for the rise in the motor current. First, degradation of the gear part can cause a rise in motor current. In this case, the motor compressing the spring deteriorated. Then, the spring is not properly compressed, so more motor current flow is required. Second, a degradation of the close shaft/spring part could cause higher motor current occurrence. If the spring's elasticity decreases, the motor cannot compress it adequately with the same current value. Consequently, a higher current is essential to compress the spring effectively.

## 5. Conclusions

In this study, we proposed the framework for GIS remanufacturing decision making based on the quantitative criterion. To validate our framework, we conducted the ALT and acquired actual GIS data from seven sensors in our laboratory. The comparison results of six regression models were reported by computing MSE, MAE, and $R^2$. In this experiment setting, Random Forest achieved the best performance against the others. In the simulation study, we calculated the increasing RUL value per cost to determine which parts are to be replaced. The process determined the most cost effective and critical parts to replace.

The framework includes sensor data collection, preprocessing, RUL regression, and cost-efficient replacement. We expect that this framework will apply not only to the electric power industry, but also to several manufacturing industries. In the robot and automobile industries, for instance, remanufacturing decisions can be made if data from sensors attached to each part can be collected. The collected data are used to predict the RUL of robots or automobiles in advance. As verified in our work, the predictive analytics results can be used for predictive maintenance, which reduces considerable maintenance costs.

The limited number of GIS types is the drawback of this paper. Although each experiment of ALT generally takes five months and requires a high cost, we are planning to conduct more ALT for data acquisition not only for the spring mechanism, but also for

the pneumatic mechanism, and for hydraulic types of circuit breakers. We will integrate frameworks for all kinds of circuit breakers in the future.

**Author Contributions:** Conceptualization, S.M., H.C., H.L.O., Y.K. and S.B.K.; methodology, S.M., H.C. and Y.K.; software, S.M. and H.C.; validation, H.C., E.K. and Y.K.; formal analysis, S.M., Y.S.C. and S.B.K.; investigation, S.M., H.C., E.K., Y.S.C., Y.K. and S.B.K.; writing—original draft preparation, S.M., H.C. and E.K.; writing—review and editing, Y.K. and S.B.K.; visualization, S.M. and Y.K.; supervision, Y.K. and S.B.K.; project administration, Y.K.; funding acquisition, Y.K. and S.B.K. All authors have read and agreed to the published version of the manuscript.

**Funding:** This research was financially supported by the Ministry of Trade, Industry and Energy, Korea, under the "Regional Innovation Cluster Development Program (R&D, P0015346)" supervised by the Korea Institute for Advancement of Technology (KIAT) and Korea Institute for Advancement of Technology (KIAT) grant funded by the Korea Government (MOTIE) (P0008691, The Competency Development Program for Industry Specialist).

**Institutional Review Board Statement:** Not applicable.

**Informed Consent Statement:** Not applicable.

**Data Availability Statement:** Due to privacy and legal restrictions, the data reported in this study cannot be made public.

**Conflicts of Interest:** The authors declare no conflict of interest.

## Abbreviations

The following abbreviations are used in this manuscript:

| | |
|---|---|
| GIS | Gas Insulated Switchgear. |
| RUL | Remaining Useful Life. |
| ALT | Accelerated Life Testing. |

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
