# Peer review of "Remanufacturing Decision-Making for Gas Insulated Switchgear with Remaining Useful Life Prediction"

_sustainability, doi:10.3390/su141912357_

Round 1

Reviewer 1 Report

Very good Article. Please check minor refuses such  as Capital Letters for Names ( such as in page 2 vector machine or kernel principal component analysis) of and similar. No other suggestions

Reviewer 2 Report

Overall, the paper is good. It provides sufficient details of information and has a proper investigation structure, flow, sufficient data and encouraging results. The paper corresponds well to the journal scope. I recommend publishing this research after minor corrections as suggested below:

1.   Page 2 line 47: what do you mean by “tree-based models can interpret the prediction results”? a brief elaboration/explanation of this sentence is recommended in the manuscript.

2.  In Figure 1: recommend to swap the location of the “validation of relation..” diagram and the table of “result of cost sum…”. Based on the explanation in line 92-93, increment of RUL per cost is set first before the validation of relation is carried out.

3.      In Figure 2: would it be possible to mark/annotate the locations of the 7 sensors in the illustration?

4.     Figure 9 is not mentioned anywhere in the manuscript. Perhaps can refer Figure 9 in page 9 line 247.

5.  Page 10 line 253: need to provide what does LSTM stand for. No information about this abbreviation is provided in the manuscript.

6.  For Figure 10: Suggest to combine them into 1 graph, since it is comparing 1000, 7000 and 14300 open/close operation curves. Two graphs give the impression that it is two different processes. If one graph cannot show the higher current of 14300, perhaps can give another graph that zoom into the 0-10seconds and 0-10current range.

7.    Please put units in all the graphs, such as the units used for current and distance.

Reviewer 3 Report

The authors have presented the Remanufacturing Decision-making for Gas Insulated Switchgear with Remaining Useful Life Prediction. Its an interesting work. The reviewer has the following comments. 

1) The authors have compared the performance six standard regression algorithms. Further, the authors stated that they explored LSTM for the same. However it results in poor performance. The comment is that whether the authors could find the scientific reason why the deep learning based regression algorithms perform poor results. It would be interesting to include these results with relevant reasons/justifications.   

2) It would be good to include the procedure to calculate/prepare RUL for any kind of system. 

3) In Equation 1, How we can define the M (maximum number of operations) for any system. 

4) Highlight the new contributions of the work with respect to the state of the art. 

5) Include details on how this system can be applied in real world application in industries. 
